# Randomized Channel Shuffling: Minimal-Overhead Backdoor Attack Detection without Clean Datasets

**Ruisi Cai[1*], Zhenyu Zhang[1*], Tianlong Chen[1], Xiaohan Chen[1], Zhangyang Wang[1]**
[1]University of Texas at Austin
{ruisi.cai,zhenyu.zhang,tianlong.chen,xiaohan.chen,atlaswang}@utexas.edu

## Abstract

Deep neural networks (DNNs) typically require massive data to train on, which is a hurdle for numerous practical domains. Facing the data shortfall, one viable option is to acquire domain-specific training data from external uncensored sources, such as open webs or third-party data collectors. However, the quality of such acquired data is often not rigorously scrutinized, and one cannot easily rule out the risk of "poisoned" examples being included in such unreliable datasets, resulting in unreliable trained models which pose potential risks to many high-stake applications. While existing options usually suffer from high computational costs or assumptions on clean data access, this paper attempts to detect backdoors for potential victim models with minimal prior knowledge. In particular, provided with a trained model, users are assumed to (1) have no prior knowledge of whether it is already poisoned, or what the target class/percentage of samples is poisoned, and (2) have no access to a clean sample set from the same training distribution, nor any trusted model trained on such clean data. To tackle this challenging scenario, we first observe the contrasting channel-level statistics between the backdoor trigger and clean image features, and consequently, how they can be differentiated by progressive channel shuffling. We then propose the randomized channel shuffling method for backdoor-targeted class detection, which requires only a few feed-forward passes. It thus incurs minimal overheads and demands no clean sample nor prior knowledge. We further explore a "full" clean data-free setting, where neither the target class detection nor the trigger recovery can access the clean data. Extensive experiments are conducted with three datasets (CIFAR-10, GTSRB, Tiny ImageNet), three architectures (AlexNet, ResNet-20, SENet-18), and three attacks (BadNets[1], clean label attack [2], and WaNet [3]). Results consistently endorse the effectiveness of our proposed technique in backdoor model detection, with margins of $0.291 \sim 0.640$ AUROC[1] over the current state-of-the-arts. Codes are available at https://github.com/VITA-Group/Random-Shuffling-BackdoorDetect.

## 1 Introduction

Numerous deep learning-based methods significantly outperform traditional methods and become useful in many critical domains. To achieve good results, the enormous deep neural networks (DNNs) require a large amount of training data. Due to the absence of large enough self-gathered datasets in many applications, many users refer to data samples collected from external sources, e.g., third-party data collection companies or other web sources. However, these external sources can become untrustworthy because the companies or the uploaders could be attackers who inject "poison" into the datasets by maliciously crafting the data samples thus resulting in unreliable models.

---

[*]Equal Contribution.
[1]*AUROC* stands for the area under the receiver operating characteristic curve.

36th Conference on Neural Information Processing Systems (NeurIPS 2022).

Recently, as one category of training-time attacks, poisoning attacks emerge as a fatal threat to deep learning applications [4, 5]. Among them, the *backdoor attack* is notably stealthy. In such an attack, the attackers will poison a small portion of training data by injecting small triggers into or applying certain transformations on them. The poisoned dataset is called a *backdoored* dataset. A backdoored dataset usually contains both poisoned samples and *benign* or clean samples that are not contaminated. Traditional backdoor attackers will also maliciously label the poisoned samples into a target class [1, 4, 6, 3] when the attackers have full control over the datasets. More recent works [2, 7] do not require this label control and craft only the raw data. For both types of backdoor attack methods, the *backdoored model*, i.e., the DNN that is trained on the backdoored dataset, will behave normally on benign inputs while classifying most malicious inputs into chosen target classes.

Due to the lack of transparency and interpretability, it is hard to determine whether an unreliable dataset or a DNN trained on it is backdoor attacked or not, how the trigger looks like under attacks, and which classes are infected. Users are left in desperate need of a reliable method that can achieve *attack detection and trigger recovery*, using their limited resources, in the absence of clean data.

Recently, many detection and trigger recovery methods have been proposed to tackle backdoor attacks [8, 9, 10, 11, 12, 13, 14, 15, 16, 17, 18, 19, 20, 17, 21]. However, most of them assume access to a hold-out dataset only containing benign samples from the training domain. Assuming such access may be unrealistic in real-world scenarios as users have no control nor prior knowledge of the training datasets. Besides, constructing a small clean set from public datasets online is often not a feasible fix either, because samples from public datasets can still possess large domain gaps from the (unknown) training data, and therefore might hardly help or even hurt the detection and/or trigger recovery.

Several methods attempted to detect backdoor attacks without assuming their access to clean data [8, 22, 23, 24]. For example, [23] trains a classification auto-encoder to distinguish backdoor data, which however incurs extra training overhead. [24] identifies backdoor samples via influence estimation while [8] analyzes the activation of the last hidden layer. In [22], the authors theoretically illustrate that the input gradient has a relatively large absolute value at the pixel positions of the stamped trigger pattern, which supplies a good indicator for distinguishing backdoor samples from benign ones. [17] proposes a data-efficient detection method by optimizing triggers for every putative target class. Yet for a potentially unreliable dataset or a model trained on it, users might prefer a more efficient and accurate method that can quickly assess the chance of being attacked, rather than exhaustive checking. Moreover, most of those existing methods remain to compromise their detection accuracy, and all still rely on clean samples for their trigger recovery step, to our best knowledge.

## 1.1 Our Contributions

In this work, we investigate a new challenging scenario: how to conduct both backdoor detection and trigger recovery without clean samples, while maintaining low overhead? Laid on the foundation of this work is our core hypothesis over the channel-level statistics of the backdoored features:

**Trigger Feature Hypothesis:** We hypothesize that *the trigger features are sparsely encoded in only a few channels, while clean image features need to be encoded across many channels for effective classification*. This is a key difference from normal data features that are presumably distributed more evenly across channels, which indicates that these two types of features might behave differently in certain situations, leading to our main technical contribution.

**Feature Differentiation:** Based on the proposed hypothesis, we devise a novel randomized channel shuffling mechanism which introduces minimal computation cost to reveal trigger features. Specifically, given an unreliable dataset and a model trained by it, we control the channel shuffling strength and observe how representations vary according to it. By observing the abnormally different variation, we can detect the target class with only a few feed-forward passes. In addition, we further utilize the proposed mechanism to separate those clean image features and trigger features. Specifically, we retrain the model using the channel shuffling operation with part of the training set for a few epochs (0.25% of total training cost and then recover triggers without clean data based on the shuffled model. The trigger can be directly recovered to the pre-identified target class thus bringing efficiency.

We summarize our novel contributions as follows:
- We discover that clean image features and trigger features are different in terms of their activation distribution in channels, as well as their sensitivity to channel re-ordering. Based on that, we propose an effective, generalizable approach based on channel shuffling to detect backdoor attacks and recover triggers with minimal overhead.

- We explore a new "head-to-toe" clean data-free setting, where neither backdoor detection nor trigger recovery requires any access to clean data from the training domain. Our channel shuffling-based detection can directly obtain the target class without model retraining. The resultant shuffled models also facilitate more effective trigger recovery, also free of clean data.
- Across various datasets (CIFAR-10, GTSRB, Tiny ImageNet), architectures (AlexNet, ResNet-20, SENet-18), and backdoor attacks (BadNets[1], clean label attack [2], and WaNet [3]), the experiment results consistently endorse the effectiveness of our technique in backdoor model detection and trigger recovery, specifically, $0.331 \sim 0.640$ higher AUROC than NC [14], and $0.291 \sim 0.543$ higher than STRIP [18]. Our trigger recovery method also achieves up to 30% improvements in attack success rate for trigger recovery than NC [14] across all scenarios.

## 2  Related Work

**Backdoor Attack and Defense.**  Several backdoor attacks on deep learning models have been proposed recently. Among them, most backdoor attacks can be classified as trigger-driven attacks. Specifically, during the training phase, to launch an attack, the adversary injects an arbitrary trigger into a small fraction of training pictures and maliciously labels them to the target class. Traditional triggers are pre-defined patterns that can be simply stamped on benign images [1, 4]. Many advanced trigger injection methods are proposed recently [6, 3] for stealthiness under both human inspection and former detection methods [14, 15, 18]. Another category of backdoor attack is clean-label attack [2, 25, 7, 26, 27, 28, 29], which is formulated without control over the labeling function. Among them, [26] targets large-scale datasets and scales up the poisoning attack method via gradient matching. Additionally, both [25] and [27] focus on improving the transferability of attack methods, while [28] utilizes meta-learning and proposes an effective and transferable poisoning method.

Earlier works detect and mitigate backdoor models based on abnormal neuron responses [8, 10, 17], feature representation[9], entropy [18], and evolution of model accuracy [12]. [19] models the distribution of Trojan attacks by meta neural analysis, [16] utilizes Deep k-NN with a focus on clean-label attack, [20] tries to learn the Universal Litmus Patterns (ULPs) from poisoned datasets.

Another category of work [14, 15, 17, 13, 11, 21] focuses on recovering the trigger and detecting backdoor models based on the properties of the recovered triggers. Utilizing clean testing images, Neural Cleanse (NC) [14] calculates minimal perturbation to cause misclassification towards every putative incorrect label. Backdoor model detection is completed by MAD outlier detection, due to the observation that the size of minimal perturbation to classify images from all classes to the target class is markedly smaller than that of other classes. However, the aforementioned methods suffer from occasional failures in detecting backdoor models. More importantly, most methods except [17] that fall in this category generally assume the defender has a clean validation set, which may not be practical. Instead, we do not assume we can obtain a clean validation set for effective detection.

We also clarify the difference between our new setting and the more classical backdoor detection setting as adopted in [30, 16, 31]. Those prior arts perform *pre-training* detection of backdoor training samples, without needing clean validation sets either. They all follow the same workflow: first detecting the poison examples in the training set, then removing them prior to training, and finally training the model using the filtered set. As the downside, to find out whether a pre-trained model has been poisoned or not, those methods also have to (re)train from scratch, which may often be too costly or impractical. In contrast, [18] and our method belong to *post-training* detection that can detect from a given pre-trained model without costly (re-)training from scratch. That is more practical when the model is already actively deployed in the field. While [18] needs a clean "reference" set, our method requires no prior-training sample selection and (re-)training (hence differing from [30, 16, 31]), while eliminating the necessity of clean samples (hence differing from [18]).

**Channel Shuffling.**  The idea of channel shuffling has mostly been used in efficient model design, intending to enable information flow between branches and consequently strengthen representation. Specifically, aiming at high efficiency, [32] and [33] utilize the channel shuffling method to mitigate the problem induced by group convolution[34, 35]. Similarly, [36] includes channel split and shuffle methods to reduce computation cost while maintaining higher segmentation accuracy. [37] includes channel shuffling to enable information communication between different sub-features. Different from the aforementioned works, [38] adopts the channel shuffling method with randomness as a regularization technique, to reduce overfitting and enhance generalization capabilities.

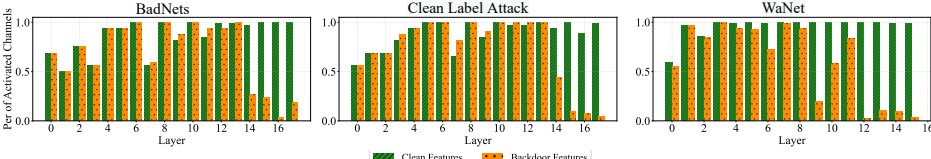

Figure 2: Green and orange columns represent the percentages of activated channels in each layer when inputting clean images and backdoor images, respectively. We empirically set the $p = 20\%$ of max activation level as the threshold and consider channels with higher activation levels than the threshold as activated.

Based on our observation that clean image features and trigger features have different channel distributions which is shown in Section 3.1, we formulated the randomized channel shuffling mechanism to reveal trigger features, aiming to detect backdoor models and recover triggers. To the best of our knowledge, we are the first to connect channel shuffling and the feature differentiation goal.

**Shortcut Features.** Several recent works [39, 40] analyzed that trigger features can serve as "shortcut" features [5], which are observed to be easier to learn than the discriminative clean image features [41]. In addition, [10] observed that a small number of neurons are activated by backdoor images but remain dormant for clean images. In contrast, [42] demonstrated the importance of the combination of channels when encoding complex and diversified image features on large datasets.

## 3 Methodology

In Section 3.1, we first present our main hypothesis on trigger features' unique characteristics compared to clean image features, alongside with experimental validation. Section 3.2 proposes our randomized channel shuffling method for backdoor model detection, as motivated by the hypothesis. Section 3.3 shows further how we can improve trigger recovery with the channel shuffling scheme.

### 3.1 Differentiating Trigger and Image Features: A Hypothesis

Authors of [5] pointed out that DNNs tend to unintendedly exploit features *unrelated* to the task of interest to achieve good performance. These "shortcut" features are usually *simpler* and *easier* to learn [41] but highly correlated to the task-related features, such as the sky background and the bird objective. It is then verified in [39, 40] that the backdoor triggers play a similar role as shortcuts exploited by the backdoored models.

Moreover, a small fraction of channels are observed to remain dormant for clean images but are activated by backdoor images [1, 10]. Based on all these observations, we conjecture that the trigger features have a key difference from clean image features in terms of channel-level activation, and raise the following **hypothesis**: *Trigger features are sparsely encoded and activated in only a few channels, while clean image features need to be encoded across many channels for effective classification.*

To validate the hypothesis, we investigate the *activation level* of a specific channel, defined as the average magnitudes of elements in its (output) feature map. In each layer, we consider channels with activation levels larger than a ratio $p$ of its highest activation level as 'activated'. We then observe how channels are activated differently for clean and backdoor images. In Figure 1, we take the second last layer in a model backdoored by BadNets [1] as an example. When the input is backdoored, only one single channel shows an ex-

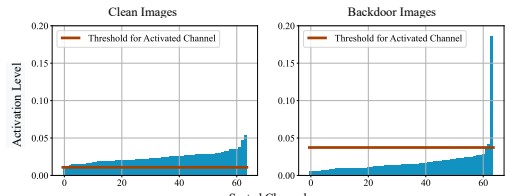

Figure 1: Blue columns represent the detailed distribution of activation levels of *the second last layer*. Red lines indicate the threshold for activated channels, 20% of the max activation level in this figure.

ceptionally high activation level, leaving all other channels "unactivated" based on our definition. In contrast, a clean input has more balanced activations across all channels. In Figure 2, we have consistent observations on more advanced backdoor methods such as clean label attack [2] and transformation-based attack, WaNet [3]. Moreover, we can see that the sparse encoding of trigger features is more obvious in the latter layers, which are known to encode more discriminate semantic features while the shallow layers usually encode histogram features.

### 3.2 Trigger Feature Exposure via Channel Shuffling

The next question is, what can be implied from the significant imbalance in the activation distribution of trigger features? Consider the encoding of a poisoned image at a latter layer in the backdoored

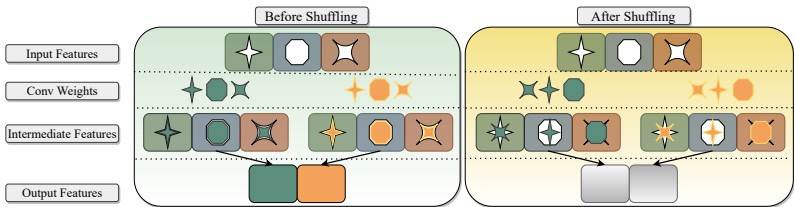

Figure 3: The overview of randomized channel shuffling operation. Before shuffling, the convolutional weights and feature maps are 'matched' in terms of channels (e.g., the input feature with a hole in the shape of a polygon is supposed to encode by the polygon weight). After channel shuffling, we deliberately induce a mismatch (the input feature with a hole in the shape of a polygon has to be encoded by the star-shaped weight), interfering with the feature encoding process and generating meaningless output features.

model, where only as few as one channel is vital with a high activation level. This channel needs to match well with convolutional weights in the next layer for the attack to succeed. However, if we deliberately create a mismatch between them, the lack of any second "vital channel" will lead to unsuccessful encoding with a high probability. In contrast, channel features of clean images are activated more evenly, leaving a lot of backup channels to substitute when a mismatch is created. Motivated by this insight, we design a randomized channel shuffling mechanism to manually induce a mismatch between the activated channels and channel weights, and propose a way to observe how much the encoding process is being affected, thus revealing trigger features.

**Randomized Channel Shuffling with Controlled Strength.** Consider the convolutional weight $\theta^l$ at the $l^{\text{th}}$ layer of shape $C_{\text{out}}^l \times C_{\text{in}}^l \times F^l \times F^l$, where $C_{\text{out}}^l$ and $C_{\text{in}}^l$ are the number of output channels and input channels, respectively; $F^l$ is the kernel size. We shuffle the order of *input channels* (the second dimension) of $\theta^l$. And the overview of randomized channel shuffling is illustrated in Figure 3.

Let $\Theta = \{\theta^1, \ldots, \theta^L\}$ denote the set of all weights in the network. We create a shuffled version of itself, $\Theta(n)$ by randomly shuffling the last $n$ convolutional layers, $n = 0, 1, ..., L-1$, since trigger features show more obvious sparsity characteristics in the latter layers according to Section 3.1. We control the strength of channel shuffling by setting different $n$ values for shuffling; the larger $n$ is, the stronger channel shuffling is applied. Then, for a given input $\boldsymbol{x}$, we calculate its representation (the output of the last convolutional layer) using $\Theta$ and $\Theta(n)$, denoted as $f_{\Theta}(\boldsymbol{x})$ and $f_{\Theta(n)}(\boldsymbol{x})$, respectively, where $f$ is the representation extraction part of a model. Finally, we compute the representation shift $s_n$ using the $\ell_2$-norm distance: $s_n(\boldsymbol{x}) = \| f_{\Theta}(\boldsymbol{x}) - f_{\Theta(n)}(\boldsymbol{x}) \|_2$.

**Detect the Class-wise Abnormal Variation.** First, for a dataset $\mathcal{D}$ with $K$ class, we split it into $K$ subsets according to the labels. For $\mathcal{D}_k$, the subset with class $k$, we calculate the standard deviation of representation shift caused by the shuffling operation on the last $n$ layers, over all samples in that subset, denoted by $y^k(n) = std_{\boldsymbol{x}_i \in \mathcal{D}_k}\{s_n(\boldsymbol{x}_i)\}$. Then, by changing $n$, we can obtain a curve $y^k[0:n]$ for class $k$ showing the variation of the standard deviation of representation shift versus different strengths of channel shuffling. Finally, We can detect the target class by observing the abnormal curve, and its rationale is rooted in our sparsity hypothesis in Section 3.1. Note that using the mean of representation shift instead of using standard deviation still works for detection, and results are provided in Appendix A3.1. We fix $n$ (maximum layer number to shuffle, starting from the top/last layer) as four, and we analyze the choice in Section 4.4.

To describe how a specific curve $y^k[0:n]$ deviates from a cluster of curves $\{y^j[0:n], j = 0, ..., K-1\}$ and how to detect an anomaly, we formulate the following two metrics:

($i$) *Distance*: We use the $\ell_1$ distance between the curve and the median to measure how far the curve is away from the cluster, formulated as $\Phi_{\text{d}} = \| y^k[0:n] - median_j\{y^j[0:n], \ j = 0, ..., K-1\} \|_1$.

($ii$) *Tendency*: First, we preprocess each curve to achieve 'zero-mean' and get $\hat{y}^j[0:n]$. Then, we calculate the standard deviation of the difference between the curve and the median. To be specific, $\Phi_{\text{t}} = std_n\{ \hat{y}^k[0:n] - median_j\{\hat{y}^j[0:n], j = 0, ..., K-1\} \}$.

Then, we adopt $\Phi_{\text{dev}} = \Phi_{\text{d}} + \lambda * \Phi_{\text{t}}$ to take both deviation in distance and tendency of a curve into account, and $\lambda$ is to balance two factors. A larger value of $\Phi_{\text{dev}}$ implies that its corresponding curve is less similar to others and more likely to be the target class. We default to set $\lambda$ as 0.01 in our experiments, and we conduct an ablation study about the effect of different $\lambda$ values in Section 4.4, showing the effectiveness of the proposed method is not sensitive to the choice. After calculating $\Phi_{\text{dev}}$ for each curve, we use *Median Absolute Deviation* to detect outliers, which is a common choice

for numerous backdoor detection methods[14, 15]. An anomaly index can be obtained for each curve, measuring the probability of the curve being an outlier and relating to the target class. By properly setting the threshold, the target class and backdoor model can be detected. In Section 4.2, we measure the detection ability of our proposed method by AUROC. In addition, our method only needs a few feed-forward passes, showing exceptional efficiency compared to existing methods. More details can be found in Appendix 4.5.

### 3.3 Improved Trigger Recovery with Shuffled Models

In this section, we improve the challenging trigger recovery by leveraging the distinctive properties of channel-level feature distributions from clean images and triggers. Specifically, we first design a training procedure with randomized channel shuffling to destroy clean image features while retaining trigger-related features. Such models perform terribly for predicting clean images, but can be easily attacked by samples with a trigger (i.e., almost perfect $\sim 100\%$ attack success rate). Then, this "impaired" model is adopted for the subsequent trigger recovery where substantial performance improvements can be observed without access to clean data.

The rationale lies in: (1) Based on our verified hypothesis in Section 3.1 - *trigger features are sparsely encoded while clean image features are encoded across multiple channels*, randomized channel shuffling alters the channel order every iteration, damages the relationship among channels, and therefore stops the model to encode meaningful clean image features. On the contrary, trigger features are more amenable to the perturbed channel orders thanks to their sparse encoding manner. (2) by disentangling trigger and clean image features, the channel-shuffled models (where clean image features are supposed to have been destroyed) should be more tractable for trigger recovery since they contain almost "pure" trigger features now. This intuition is echoed by recent findings in [43].

**Training with Randomized Channel Shuffling.** Given a backdoor model $\mathcal{G}$, the randomized channel shuffling is applied at each training iteration and produces its shuffled variant $\tilde{\mathcal{G}}$ in the end. We find that a few training epochs (e.g., 5 epochs) on a subset (e.g., $10\%$) of the original training set, are sufficient to generate a high quality $\tilde{\mathcal{G}}$ with low clean testing accuracy and high attack success rate.

**Enhanced Backdoor Trigger Recovery without Clean Data.** In the backdoor model $\tilde{\mathcal{G}}$, the presence of clean image features and the stealthiness of trigger features make it hard to directly activate trigger features when users lack knowledge of the trigger pattern. Previous approaches typically rely on clean images to activate clean image features, helping to reveal trigger features based on the connection between trigger features and clean image features[14]. In contrast, $\tilde{\mathcal{G}}$ has enriched trigger features with suppressed clean image features, making the trigger feature exposure much easier, which also removes its dependency on clean data as the reference. Therefore, on top of previous approaches, we improve its trigger recovery performance by utilizing $\tilde{\mathcal{G}}$. In practice, we use noise images drawn from the standard Gaussian distribution to substitute clean data. And the recovered triggers have similar high quality as with clean data, as shown in Section 4.3. Additionally, we introduce an enhanced objective function as depicted below:

$$\mathcal{L}(\boldsymbol{x}, \mathtt{M}, \Delta, y_t) \quad = \overbrace{\mathcal{L}_{\mathrm{XE}}(\mathcal{G}(\boldsymbol{x}'), y_t)}^{\text{Classical Term [14]}} + \overbrace{\mathcal{L}_{\mathrm{XE}}(\tilde{\mathcal{G}}(\boldsymbol{x}'), y_t)}^{\text{Our Term with Shuffled Models}} \quad , \tag{1}$$

where $\boldsymbol{x}' = \boldsymbol{x} \odot (\mathbf{1} - \mathtt{M}) + \mathtt{M} \odot \Delta$, $\boldsymbol{x}$ is the randomly generated noise image and $y_t$ denotes the target label detected in our previous stage (Section 3.2). $\mathcal{L}_{\mathrm{XE}}(\cdot)$ refers to a cross-entropy loss, $\Delta$ represents a recovered trigger pattern, and the mask $\mathtt{M}$ characterizes the region that the trigger stamped on. Here $\mathtt{M}$ is a binary mask, belonging to $\{0, 1\}^{d \times d}$, where $d \times d$ denotes the resolution of input images.

## 4 Experiments

### 4.1 Implementation Details

**Trojan Details.**   We consider three representative backdoor attacks from two categories in our experiments, including ($i$) traditional backdoor attacks, *i.e.*, BadNets [1] and WaNet [3]; ($ii$) clean label attack (CLA) [44]. Among them, WaNet is more advanced which is designed to be more stealthy to human inspection [3]. We have also compared with [45] in Appendix A3.2.

For BadNets [1], we adopt $5 \times 5$ gray-scale triggers. And $10\%$ of the full training samples are injected with the trigger. More details about the trigger injection and the trigger pattern are provided in Appendix A1.1. And for CLA [2], we craft all training images from the target class and stamp the gray-scale trigger. We follow the default hyperparameter configurations in [3], in which $10\%$ of total

training images are warped. Also, it is worth mentioning that we adopt the "noise mode" of WaNet, which is reported to be more stealthy and challenging.

**Datasets and Architectures.** We evaluate our randomized channel shuffling method across various backdoor attacks, datasets and architectures with a total of seven combinations, *i.e.*, {BadNets, CIFAR-10, ResNet-20}, {BadNets, CIFAR-10, AlexNet}, {BadNets, CIFAR-10, SENet-18}, {BadNets, GTSRB, ResNet-20}, {BadNets, Tiny ImageNet, ResNet-18}, {CLA, CIFAR-10, ResNet-20} and {WaNet, CIFAR-10, ResNet-18}. For each combination, we train 25 backdoor and 25 benign models via independent runs. Training details are collected in Table 1.

Table 1: Detailed training configurations of the backdoor injection procedure.

| Attacks | # Epoch | Learning Rate | Optimizer | Batch Size | Target Label | Poison Ratio | $s$ | $k$ | Cross Ratio |
|---|---|---|---|---|---|---|---|---|---|
| BadNets [1] | 200 | 0.1 | SGD | 128 | "0" | 0.1 | - | - | - |
| Clean Label Attack [2] | 200 | 0.1 | SGD | 128 | "0" | 0.1 | - | - | - |
| WaNet [3] | 200 | 0.01 | SGD | 128 | "0" | 0.1 | 0.8 | 6 | 2 |

**Detection and Evaluation Details.** ▷ *Detection.* To examine the reliability of the training dataset with N classes, we first divide it into N subsets based on their label. For each subset, we then feed them into the target model as well as its randomly shuffled variant, and compute the associated representation shifts over different numbers of shuffled layers. Based on our observations in Section 3.1, the last few layers mainly encode discriminate features and therefore they are used in our detection. In our implementation, we only shuffle the channel order within the last four layers and generate feature sensitivity curves as $\{y^k[n], k = 0, ..., N - 1, n = 0, 1, 2, 3\}$. $y^i[j]$ denotes the standard deviation of representation shift from the subset of class $i$ after randomly shuffling the last $j$ convolutional layer. Note that the range of the number of layers used for our detection plays a minor role in the achievable performance, as evidenced in Section 4.4. Lastly, we utilize the metric $\Phi_{\text{dev}}$ shown in Section 3.2 to measure whether there exists an outline curve significantly different from other curves. An abnormal curve usually indicates: $i)$ the dataset and model contain backdoor triggers; $ii)$ its corresponding class is likely to be the backdoor target class. ▷ *Evaluation.* Similar to [17], we use *area under the receiver operating characteristic curve* (AUROC) to measure the backdoor model detection's performance. More experimental details are provided in Appendix A2.

### 4.2 Backdoor Detection by Leveraging Abnormal Variation

**Superior Performance across Diverse Datasets and Architectures.** We verify the effectiveness of the proposed detection algorithm with the randomized channel shuffling method on seven representative combinations of datasets, network architectures, and attack methods. Detection's ROC results are presented in Figure 4, where orange, green, and blue curves are generated by Ours, Neural Cleanse [14], and STRIP [18], respectively. Consistent performance improvements are observed: ① *Across Architectures*: Compared with NC and STRIP, our method achieves an improvement of AUROC values on CIFAR-10 with BadNets by {0.331, 0.375, 0.339} and {0.541, 0.543, 0.306} for architectures {ResNet-20, AlexNet, SENet-18}, respectively. ② *Across Datasets*: On GTSRB and Tiny ImageNet, our approach consistently outperform NC by a substantial margin of 0.539 and 0.388 AUROC values, with 0.293 and 0.457 improvements compared to STRIP. ③ *Across Attack Methods*: Even for more advanced attack methods CLA and WaNet, our methods still gain 0.362, 0.291, and 0.64, 0.512 AUROC improvements on CIFAR-10 against NC and STRIP.

For further evaluation, we calculate the percentage of successfully detected models with a fixed threshold. And we choose the threshold that maximize the total detection accuracy, i.e., (*# correctly detected benign models and backdoor models*) / (*# total models*). As shown in Table 2, in each combination of {dataset, architecture, attack method}, our detection method consistently outperforms other competitive approaches by up to 37.72% detection accuracy.

**Feature Sensitivity Curves and the Distribution of Indicator $\Phi_{\text{dev}}$.** We present the feature sensitivity curves and the distribution of indicator $\Phi_{\text{dev}}$ in Figure 5 and Figure 6, respectively. Without loss of generality, for each attack, we randomly select one model for further analysis. Each point in Figure 5 denotes the standard deviation of feature shifts between the original and randomly shuffled models, given a subset of input images from a certain class. After we vary the number of shuffled layers, a feature-sensitive curve can be drawn. From the results in Figure 5, we see the standard deviation of feature shifts on the target class's samples appears as a distinctive pattern (red curves), especially for the first few points, which lies the foundation of our detection algorithm.

Table 2: Comparisons between ours and classic detection approaches. The reported numbers represent (*# correctly detected models*) / (*# models*). Note that "'CLA' stands for the clean label attack.

| Attack | Dataset | Arch | Neural Cleanse | | | STRIP | | | Ours | | |
|--------|---------|------|--------|----------|-------|--------|----------|-------|--------|----------|-------|
| | | | benign | backdoor | AUROC | benign | backdoor | AUROC | benign | backdoor | AUROC |
| BadNets | CIFAR-10 | ResNet-20 | 14/25 | 20/25 | 0.669 | 18/25 | 8/25 | 0.459 | 25/25 | 25/25 | 1.000 |
| | | AlexNet | 19/25 | 13/25 | 0.622 | 16/25 | 11/25 | 0.454 | 24/25 | 25/25 | 0.997 |
| | | SENet-18 | 15/25 | 18/25 | 0.578 | 12/25 | 13/25 | 0.611 | 23/25 | 21/25 | 0.917 |
| | GTSRB | ResNet-20 | 24/25 | 2/25 | 0.341 | 12/25 | 21/25 | 0.587 | 23/25 | 21/25 | 0.880 |
| | Tiny ImageNet | ResNet-18 | 19/25 | 15/25 | 0.612 | 11/25 | 15/25 | 0.543 | 25/25 | 25/25 | 1.000 |
| CLA | CIFAR-10 | ResNet-20 | 14/25 | 20/25 | 0.635 | 20/25 | 15/25 | 0.706 | 24/25 | 25/25 | 0.997 |
| WaNet | CIFAR-10 | ResNet-18 | 25/25 | 1/25 | 0.314 | 25/25 | 1/25 | 0.442 | 21/25 | 23/25 | 0.954 |
| | **Total** | | 130/175 | 89/150 | - | 114/175 | 84/175 | - | 165/175 | 165/175 | - |

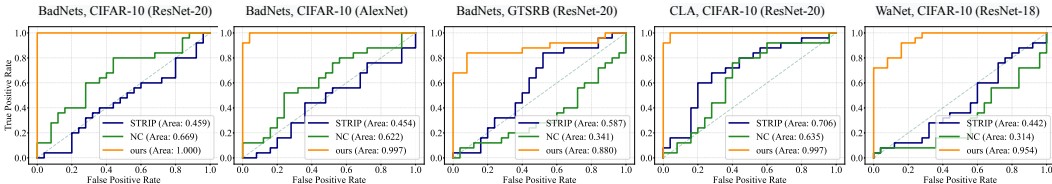

Figure 4: Performance of existing and our detection approaches, measured by ROC and related AUROC values.

On top of it, we compute the indicator $\Phi_{dev}$ to quantitatively measure the deviation from the majority of curves. Then, in order to locate the abnormal curve, the MAD outlier detection [46] is performed to $\Phi_{dev}$. Figure 6 shows that the $\Phi_{dev}$ of the target label tends to have significantly large values compared to the ones of benign labels, which allows the target label to be easily identified.

## 4.3 Improved Triggers Recovery by Models Trained with Randomized Channel Shuffling

Table 3: TA (%) and ASR (%) after randomly shuffling channels on various dataset-architecture combinations.

| Attack | Dataset | Arch | Type | TA | ASR |
|--------|---------|------|------|-----|-----|
| BadNets | CIFAR-10 | ResNet-20 | Orig. | $91.4 \pm 0.5\%$ | $100.0 \pm 0.0\%$ |
| | | | Shuf. | $16.2 \pm 7.3\%$ | $100.0 \pm 0.0\%$ |
| | | AlexNet | Orig. | $85.7 \pm 0.5\%$ | $99.9 \pm 0.1\%$ |
| | | | Shuf. | $45.4 \pm 6.7\%$ | $95.4 \pm 4.6\%$ |
| | GTSRB | ResNet-20 | Orig. | $98.5 \pm 0.5\%$ | $100.0 \pm 0.0\%$ |
| | | | Shuf. | $25.9 \pm 17.4\%$ | $98.4 \pm 1.6\%$ |
| CLA | CIFAR-10 | ResNet-20 | Orig. | $83.3 \pm 0.4\%$ | $90.1 \pm 8.0\%$ |
| | | | Shuf. | $14.36 \pm 6.1\%$ | $99.2 \pm 0.8\%$ |
| WaNet | CIFAR-10 | ResNet-18 | Orig. | $93.1 \pm 0.4\%$ | $99.2 \pm 0.8\%$ |
| | | | Shuf. | $10.2 \pm 2.3\%$ | $97.9 \pm 1.8\%$ |

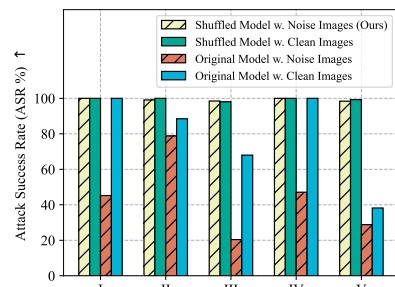

Figure 7: Attack Success Rate (ASR %) of recovered triggers from "original model" ($\mathcal{G}$) and "shuffled model" ($\tilde{\mathcal{G}}$), using clean and randomly generated noise images as inputs respectively. I, II, III, IV and V represent different backdoor configurations, i.e., {BadNets, CIFAR-10, ResNet-20}, {BadNets, CIFAR-10, AlexNet}, {BadNets, GTSRB, ResNet-20}, {CLA, CIFAR-10, ResNet-20} and {WaNet, CIFAR-10, ResNet-18}.

**Training with Randomized Channel Shuffling Leads to Low TA and High ASR.** We first investigate how the proposed training procedure with randomized channel shuffling shapes the behaviors of backdoor models. Results of TA and ASR are collected in Table 3, where "Orig." and "Shuf." indicate the original and shuffled models. Our proposal effectively produces models with nearly unimpaired ASR (e.g., $90 \sim 100\%$) but poor TA which closes the level of random guessing. Such consistent findings can be drawn from all {dataset, architecture} combinations. It again supports our claim that randomized channel shuffling tends to destroy clean image features while preserving trigger features.

**Superior Trigger Recovery.** To efficiently generate high-quality triggers, it requires two key factors: (1) detected backdoor models and identified target labels, obtained from our detection algorithms; (2) shuffled trained models that have rich trigger features but less clean image features. As evidenced in Figure 7, while previous methods can achieve acceptable results when assuming access to clean

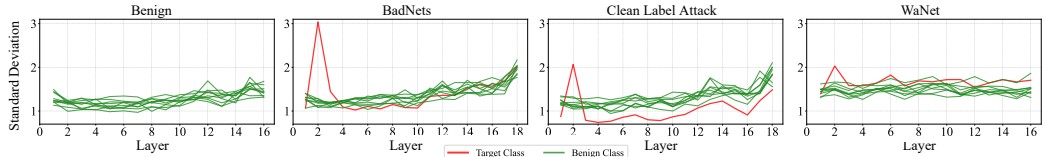

Figure 5: Standard deviation of feature shifts after applying randomized channel shuffling operation over the number of shuffled convolutional layers. Red curves are generated from subsets of the target class, while green curves are calculated from subsets of benign classes.

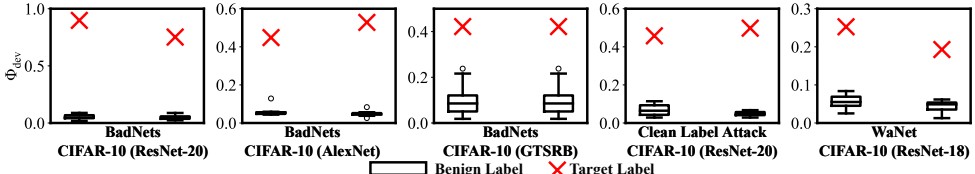

Figure 6: The $\Phi_{dev}$ distribution for backdoor models. Boxplots show the means and variances of $\Phi_{dev}$ of curves related to benign classes. The red crosses stands for $\Phi_{dev}$s of curves for target classes. A larger value of $\Phi_{dev}$ implies that its corresponding curve is less similar to others, and thus more likely to be the target class.

images, the performances drop tremendously when only using noise images. In contrast, even with noise images, our method can recover as high-quality triggers as with clean images, significantly surpassing the original models. Details are provided in Table 4. We perform experiments on 8 2080Ti GPUs. For fair evaluation, we binarize the masks and make their $\ell_1$ norms equal the ground-truth values ($5 \times 5$). Then, we calculate the attack success rates with the binary masks.

Table 4: Detailed configurations of trigger recovery methods.

| Stage | # Epoch | Learning Rate | Optimizer | Batch Size | Num. of Trainset | Num. of Cleanset |
|---|---|---|---|---|---|---|
| **Retrain** | 5 | 0.1 | SGD | 128 | 5120 | - |
| **Recovery** | 200 | 0.1 | SGD | 128 | - | 1000 |

## 4.4 Ablation Study

**Different Trigger Types in BadNets.** Instead of fixing the trigger at the lower right corner of the image, we change the trigger location to the upper left side, and randomly choose the location of the trigger to generate two types of poisoned datasets with different stamping location strategies. We further change the gray-scale trigger into the RGB trigger. We use ResNet-20 and CIFAR-10 in the above settings. The AUROC values are reported in Table 5, which demonstrate the effectiveness of the proposed randomized channel shuffling method across multiple trigger positions and types.

Table 5: The detection performance (AUROC) of ablation studies.

| Ablation Type | Different Trigger Types | | | Multiple Target Classes | | | | Partial Backdoor |
|---|---|---|---|---|---|---|---|---|
| | Bottom Left | Random Location | RGB Trigger | 2 Targets | 3 Targets | 4 Targets | 5 Targets | |
| **AUROC** | 1.000 | 1.000 | 1.000 | 0.978 | 0.981 | 0.902 | 0.770 | 0.945 |

**Multiple Target Classes.** We further consider the scenario where multiple backdoors are inserted into the dataset, with more than one class as the target class. Using CIFAR-10 as the dataset which has ten classes, ResNet-20 as architecture, and BadNets as the attack, we find that our method is still effective when multiple target classes exist but target classes still being the minority compared to the benign classes. AUROC values are reported in Table 5. Detection performances remain acceptable when fewer than 5 classes are infected. But AUROC drops significantly when 5 out of 10 classes are the target classes. More visualization results are provided in Figure 8.

**Source Label-Specific (Partial) Backdoor.** A 'partial' backdoor is activated only when inputs are from specific source labels, which is more stealthy than a non-partial backdoor. We then verify the effectiveness of our method under this scenario with BadNets[1] as the attack, ResNet-20 as the model architecture, and CIFAR-10 as the dataset. The results are provided in Table 5, showing the effectiveness of the proposed method in the partial backdoor scenario.

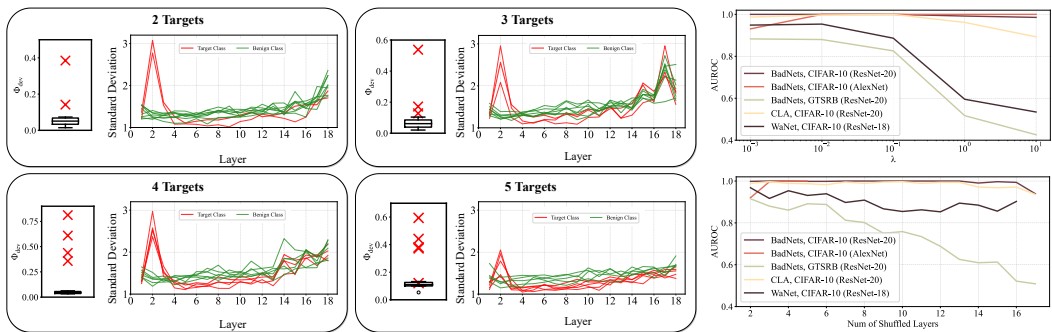

Figure 8: The *left* two columns provide visualization results when multiple backdoor target classes exist. For each attack scenario, the distribution of $\Phi_{dev}$ is collected in the *left* sub-figure, and its standard deviation of representation shifts is reported in the *right* sub-figure. The *upper right* figure provides AUROC values over different $\lambda$, the *bottom right* figure shows AUROC values over the number of shuffled layers.

**Choice of Hyper-Parameters.** As is mentioned in Section 3.2, $\Phi_{dev} = \Phi_d + \lambda * \Phi_t$, where $\lambda$ is to balance two terms. Figure 8 shows the variation of AUROC when changing $\lambda$. In addition, we also verify how the number of layers included for detection affects the detection performance for different attacks. According to Figure 8, curves of AUROC remain high when only the last several layers are included. But they gradually decrease when including more layers, since the standard deviation of representation shift from the target class becomes mixed with benign ones, affecting the performance.

## 4.5 Efficiency of Proposed Detection Method

Compared to existing methods, we claim that our proposed detection method is more efficient. We approximate the FLOPs of the back-propagation to be twice that of forwarding propagation following[47], and provide the total FLOPs needed for detection in Table 6. For CIFAR-10, NC and STRIP need $7.5 \times 10^5$ and 15.6 times more FLOPs than ours, respectively. For GTSRB, NC and STRIP need $1.9 \times 10^5$ and 3.9 times more FLOPs than ours, respectively.

## 4.6 Efficiency of Proposed Trigger Recovery Method

We claim that our trigger recovery method is also efficient since only 5 epochs and $10\%$ of the original training set are used. This is motivated by the fact that the ASR and TA saturate after a few epochs if a model is shuffled and retrained on a full dataset according to the training curve in Figure 9.

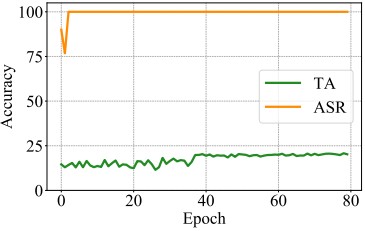

Figure 9: Test Accuracy (TA %) and Attack Success Rate (ASR %) when retraining with randomized channel shuffling on CIFAR-10 attacked by Bad-Nets, based on ResNet-20.

Table 6: The approximate of FLOPs($\times 10^{10}$) need for detection.

| Attack | Dataset | Arch | Neural Cleanse | STRIP | Ours |
|---|---|---|---|---|---|
| BadNets | CIFAR-10 | ResNet-20 | $5.04 \times 10^6$ | $1.05 \times 10^2$ | 6.72 |
| | | AlexNet | $7.73 \times 10^6$ | $1.61 \times 10^2$ | 10.30 |
| | GTSRB | ResNet-20 | $5.04 \times 10^6$ | $1.05 \times 10^2$ | 26.88 |
| CLA | CIFAR-10 | ResNet-20 | $5.04 \times 10^6$ | $1.05 \times 10^2$ | 6.72 |
| WaNet | CIFAR-10 | ResNet-18 | $6.84 \times 10^7$ | $1.43 \times 10^3$ | 91.2 |

## 5 Conclusion

In this paper, we introduce and investigate a challenging backdoor detection setup without access to clean datasets. An efficient detection algorithm is proposed by contrasting representations shifts before and after randomized channel shuffling of clean and trigger features. Moreover, by leveraging our shuffled backdoor models, the performance of trigger recovery is substantially boosted. Extensive experiments validate the effectiveness of our proposal.

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
