# OpenReview forum: "Randomized Channel Shuffling: Minimal-Overhead Backdoor Attack Detection without Clean Datasets"
_NeurIPS.cc/2022/Conference — NeurIPS 2022 Accept_

### Official Review · Reviewer_PjHg · 2022-06-28

**Rating:** 5
**Confidence:** 4
**Soundness:** 3 good
**Presentation:** 2 fair
**Contribution:** 2 fair

**Summary:**

This paper proposes a backdoor detection method based on shuffling convolutional filters and assessing the resulting shift in deep representations. Experiments are presented on two datasets, two architectures, and three backdoor attacks.

**Questions:**

- How does this method work on recent neural network architectures?
- How does this method fair on datasets that are larger, i.e. ImageNet?
- How does this defense perform against stronger poisoning attacks?

**Limitations:**

The limitations are described in Section A4. There is very little description of ways in which this work on data security and model vulnerability is applicable (or not applicable) to modern architectures and real world data. In other words, a paper that purports to defend against any kind of attack should probably explain when and where this defense will work in practice. In my opinion, discussing limitations is critical for work on security and privacy.

**Strengths And Weaknesses:**

Strengths:
- To the best of my knowledge, this is a novel approach wherein filter shuffling changes the representation.
- There is an interesting method, and the results support the hypothesis in Section 3.

Weaknesses:
- Novelty: The abstract makes big claims (in bold) that may not be true. There are existing defense techniques that do not require clean data. Some papers that should be cited and methods to compare with the proposed method (others may exist):
    + Steinhardt, Jacob, Pang Wei W. Koh, and Percy S. Liang. "Certified defenses for data poisoning attacks." Advances in neural information processing systems 30 (2017). (https://arxiv.org/pdf/1706.03691.pdf)
    + Paudice, Andrea, et al. "Detection of adversarial training examples in poisoning attacks through anomaly detection." arXiv preprint arXiv:1802.03041 (2018). (https://arxiv.org/pdf/1802.03041.pdf)
    + Peri, Neehar, et al. "Deep k-nn defense against clean-label data poisoning attacks." European Conference on Computer Vision. Springer, Cham, 2020. (https://arxiv.org/pdf/1909.13374.pdf)
- This work considers relatively outdated architectures and weak attacks. Some stronger poisoning and backdoor attacks (more exist):
    + Geiping, Jonas, et al. "Witches' brew: Industrial scale data poisoning via gradient matching." arXiv preprint arXiv:2009.02276 (2020).(https://arxiv.org/pdf/2009.02276.pdf)
    + Zhu, Chen, et al. "Transferable clean-label poisoning attacks on deep neural nets." International Conference on Machine Learning. PMLR, 2019. (http://proceedings.mlr.press/v97/zhu19a/zhu19a.pdf)
    + Aghakhani, Hojjat, et al. "Bullseye polytope: A scalable clean-label poisoning attack with improved transferability." 2021 IEEE European Symposium on Security and Privacy (EuroS&P). IEEE, 2021.
    + Saha, Aniruddha, Akshayvarun Subramanya, and Hamed Pirsiavash. "Hidden trigger backdoor attacks." Proceedings of the AAAI conference on artificial intelligence. Vol. 34. No. 07. 2020.
- Missing related work. Some of the papers above have not even been cited. Some other papers probably should be cited. I'll point to a couple survey papers here:
    + Ahmed, Ibrahim M., and Manar Younis Kashmoola. "Threats on Machine Learning Technique by Data Poisoning Attack: A Survey." International Conference on Advances in Cyber Security. Springer, Singapore, 2021.
    + Goldblum, Micah, et al. "Dataset security for machine learning: Data poisoning, backdoor attacks, and defenses." IEEE Transactions on Pattern Analysis and Machine Intelligence (2022).
- The writing is often unclear with many grammatical errors. Some examples:
    + Line 67 says that the paper will focus on backdoor attacks. Presumably, this is as opposed to poisoning attacks without a trigger. However, one of the three attacks is CLA, which has no trigger in the original method.
    + Line 30 says "poinson."
    + Line 97: "Specifically, we retrain the model using the channel shuffling operation, and then recover trigger based on the shuffle model..." Perhaps this should be "... recover the trigger..." or "... recover triggers ...".
    + Line 52 in the appendix says "We will future extend to this case."

Overall, I think there are interesting experiments, but there are problematic claims and missing relevant works/benchmarks. In my opinion, this paper is well below the threshold of acceptance in the state it is in. Short of showing that the proposed method does, in fact, beat the state of the art in poisoning detection/defense and better contextualized this method in the existing work, I recommend rejection.


----

After reading the authors' response and the updated submission (August 7):

My major concerned have been largely addressed. Thanks to much improved clarity around the setting, better contextualization with existing work, and additional experimental results, I can say this paper has much improved. I have increased my score accordingly and followed up with the comments below.

As a minor note, the discussion on limitations is still lacking.

---

> ### Author Response · Authors · 2022-08-02
> **Response to Reviewer PjHg [Cons 1 and 2]**
>
> **[Cons 1: claim validity w.r.t. missing literature]** Thanks for suggesting those references, which are indeed relevant to our methods in a broad sense, as they “detect the existence of backdoor samples without assuming clean data access”.  We have added those citations. However, those methods actually fall into a different setting from our proposed one. We apologize for not having clarified the differences clearly enough in the original version, and we see how that caused your confusion. We now clarify and compare them as follows.
>
> - **Setting 1** *(pre-training detection of backdoored training samples, without clean validation set)*, to which all suggested references [1,2,3] all belong. In particular, to detect backdoor samples in a given training set, [1,2] used an outlier detector while [3] leveraged the feature representation neighborhood (practically when defending feature collision attacks, it needs a clean set too). However, they all follow the same workflow: first detecting the poison examples in the training set, then removing them prior to training, and finally training the model using the filtered set. On the downside, to find out whether a pre-trained model has been backdoored or not, those methods also have to (re)train-from-scratch, which may often be too costly or impractical.
>
> - **Setting 2** *(post-training detection of backdoored testing samples, with clean validation set)*, to which our main comparison method STRIP [4] belongs. This is the setting that we were originally motivated from: a pre-trained model already being given, performing run-time backdoor detection.  That avoids costly re-training and is more practical when the (backdoored or benign) model is already actively deployed in the field. However, the main downside of [4] and other similar methods lies in their demand for a clean “reference” set.
>
> - **Setting 3** *(post-training detection of backdoored models/training sets, without clean validation set)*, **which our proposed method falls under**. Ours requires no prior-training sample selection and (re-)training (hence differing from setting 1) while avoiding the necessity of clean sample sets (hence differing from setting 2). As the other reviewer oFCu pointed out, one merit of this work is “in generating a very lightweight baseline that requires no repeated training” - which exactly poised our work against Setting 1.
>
> Hence, we find that our claim remains valid despite several missing references (again we apologize for not having included and discussed them initially), and our experimental results are strong to support our claims. We hence see our claim as precise and not over-bold.
>
> [1] Certified defenses for data poisoning attacks.
>
> [2] Detection of adversarial training examples in poisoning attacks through anomaly detection.
>
> [3] Deep k-nn defense against clean-label data poisoning attacks.
>
> [4]  STRIP: A Defense Against Trojan Attacks on Deep Neural Networks
>
> **[Cons 2: Effectiveness under recent attack methods, model architectures, and larger dataset]**
>
> - **Recent attack method**: To our best knowledge, WaNet [5] (2021 ICLR) is one of the most recent attacks. Our method detects models attacked by WaNet successfully, showing effectiveness.
>
> - **Recent model architectures**: We test our method on SeNet [6]  and provide results in Table S6, showing our method is applicable to recent model architectures.
>
> - **Larger dataset**: We additionally validate the effectiveness of the proposed method on Tiny-ImageNet, and the result is provided in Table S6, showing the effectiveness on a larger dataset. We will include results on ImageNet in the final version.
>
> Table S6: Detection performance on TinyImageNet. The reported numbers represent (# correctly detected models / # models).
> | Settings | benign | backdoor | AUROC |
> | :--------: |:--------: | :--------: | :--------: |
> | BadNets, TinyImageNet, ResNet-18 | 25/25 | 25/25 | 1.000 |
> | BadNets, CIFAR-10, SeNet-18  | 23/25 | 21/25 | 0.917 |
>
> [5] WaNet - Imperceptible Warping-based Backdoor Attack
>
> [6] Squeeze-and-Excitation Networks

---

> ### Author Response · Authors · 2022-08-02
> **Response to Reviewer PjHg [Cons 3 and 4]**
>
> **[Cons 3: more related work]**  Thanks for pointing it out! We have cited the missing related works[4-10] in the revised draft. Specifically, we discussed methods [1,2,3] in Section 2:
>
> *“We also clarify the difference between our new setting, and the more classical backdoor detection setting as adopted in [1,2,3]. Those prior arts perform $\textit{pre-training}$ detection of backdoor training samples, without needing clean validation sets either. They all follow the same workflow: first detecting the poison examples in the training set, then removing them prior to training, and finally training the model using the filtered set. As the downside, to find out whether a pre-trained model has been poisoned or not, those methods also have to (re)train from scratch, which may often be too costly or impractical. In contrast, [4] and our method belong to $\textit{post-training}$ detection that can detect from a given pre-trained model without costly (re-)training from scratch. That is more practical when the model is already actively deployed in the field. While [4] needs a clean ``reference” set, our method requires no prior-training sample selection and (re-)training (hence differing from [1,2,3]), while eliminating the necessity of clean samples (hence differing from [4]).”*
>
> Also, we added more advanced clean-label attack methods[7-10] in Section 2: *“Among them, [7] targeted on large scale dataset and scale up the poisoning attack method via gradient matching. Additionally, both [8] and [9] focused on improving the transferability of the attack method, while [10] utilized meta-learning and proposed an effective and transferable poisoning method.”*
>
> [7] Witches' brew: Industrial scale data poisoning via gradient matching
>
> [8] Transferable clean-label poisoning attacks on deep neural nets.
>
> [9] Bullseye polytope: A scalable clean-label poisoning attack with improved transferability.
>
> [10] Metapoison: Practical general-purpose clean-label data poisoning.
>
> **[Cons 4: writing quality]** 1) Sorry for the confusion. In terms of clean label attack, we follow the trigger version of [11] instead of the original one. We have clarified it in the revised draft. 2) replace *‘poinson’* with *‘poison’*. 3) replace *‘recover trigger’* with *‘recover triggers’*. 4) replace with *‘We will extend our method to this case in the future.’*
>
> [11] Label-Consistent Backdoor Attacks

---

> ### Author Response · Authors · 2022-08-06
> **Sincerely expecting further discussions from Reviewer PjHg**
>
> Dear Reviewer PjHg,
>
> Thank you very much for sparing your time to review our paper. In the posted response, we have tried our best to (1) clarify our setting, (2) include and discuss more related works, (3) conduct extra suggested experiments, and (4) improve writing.
>
> Since the author-reviewer discussion period has started for a few days, we would appreciate it if you could check our response to your review comments soon.
>
> We hope that you can find our effortful response convincing. If you have additional comments, please feel free to let us know. We will try our best to address them.
>
> Many thanks for your time and efforts.
>
> Best wishes,
>
> Authors

---

> > ### Comment · Reviewer_PjHg · 2022-08-07
> > **Follow up**
> >
> > Thank you for the detailed response and for making the corresponding changes in the paper. I am humbled by the clarity around the threat model and I appreciate the inclusion of a newer attack and a better discussion of related work.
> >
> > I am still slightly concerned that backdoor papers should compare on larger datasets with more up-to-date architectures. While there is one small table with SE-Net in the author response, it isn't yet in the paper. Furthermore, the argument that this defense doesn't require retraining while outlier detection methods do is interesting, but under-discussed in the paper. I think it is important to further compare to methods like [1, 2, 3] and show that they take much more compute (this could be measured a number of ways including runtime). If those techniques are stronger defenses at the cost of compute, it should be mentioned, if they are weaker (and incur a larger cost) the proposed method would be even more attractive.
> >
> > After reading the updated draft and all the reviews/responses, I have updated my score -- I recommend accepting this paper.
> >
> > [1] Certified defenses for data poisoning attacks.
> > [2] Detection of adversarial training examples in poisoning attacks through anomaly detection.
> > [3] Deep k-nn defense against clean-label data poisoning attacks.

---

> > > ### Author Response · Authors · 2022-08-09
> > > **Thanks to reviewer PjHg**
> > >
> > > Dear Reviewer PjHg,
> > >
> > > We appreciate the valuable advice and the positive assessment.
> > >
> > > In the revised version, we have made several revisions:
> > > - Provide results of the larger dataset and more recent architectures (line 310-314 & table 1);
> > > - Discuss our method does not need retraining (line 137-147);
> > > - Further compare our method with [1,2,3] (supplementary A3.3);
> > > - Discuss our limitations in supplementary A4.
> > >
> > > Thanks again for your time and efforts.
> > >
> > > Best wishes,
> > >
> > > Authors

---

### Official Review · Reviewer_oFCu · 2022-07-04

**Rating:** 7
**Confidence:** 4
**Soundness:** 3 good
**Presentation:** 4 excellent
**Contribution:** 4 excellent

**Summary:**

This paper casts a new "self-supervised" backdoor detection setting that was not explored by prior arts: there is no prior knowledge nor control of the training data quality; and no access to a comparable clean sample set. The authors presented several interesting observations and motivated an efficient backdoor detection approach for this new setting.

**Questions:**

Please check and answer the questions in "Weaknesses" section and the following questions:

- How sensitive the algorithm is to the threshold hyperparameter?
- Why use the last layers, and what if applying the shuffling to the first few or a random subset of layers?
- Why WaNet is missing from Table 3 and Table 4?
- Writing quality is overall not high, except for the abstract which appears to be well polished. The same polishing must be done towards the remaining paper.
- “Feature Disentanglement” is used in this paper to refer to a completely different meaning from its common context (in GANs etc). Using another term such as differentiation is perhaps better.

**Limitations:**

Social impacts have been properly discussed.

**Strengths And Weaknesses:**

Strengths:

+ This paper seems to identify a practically meaningful and novel backdoor setting, as ubiquitous and non-certified data sources are used for training big models nowadays (yet, potentially suffering over-rigor, see weakness).
+ The work has an empirical nature and has no theoretical foundation. However, the authors managed to dig some reasonable foundation to support their hypothesis, both from prior literature and from proof-of-concept experiments, so this is appreciated. The storytelling in Sections 3.1 and 3.2 help readers understand a lot.
+ Fundamentally, their observation is rooted in the “shortcut” property of backdoor features (noticed by prior work), which could be why the backdoor features are more sparsely encoded in channels. However, leveraging this property via shuffling is an original idea, and Figure 2 gives out why. Just a thought: would this shuffling possibly be made into a more general purpose to probe a trained neural network’s stability, with little overhead?
+ Empirical results appear to be extensive. Two datasets, two architectures, and three attacks are evaluated, including the advanced WaNet. The performance gains of 0.291 ∼ 0.640 AUROC over SOTA are impressive, considering its little overhead. Both backdoor detection and trigger recovery are considered.

Weaknesses:

My main concern is whether the authors’ proposed new setting is over-restricted or unrealistic at all. Specifically, the authors verbally claim that online public datasets are not helpful because in domain-specific applications they can be out-of-distribution. They might be true, but never manifested in practical experiments, so I’m not convinced. For example, what if one just uses a clean set from ImageNet as clean data surrogate to detect backdoors? Please justify with more concrete evidence or experiments.

Even admitting this setting, I feel more methods are available. For example, what if first performing self-supervised pre-training, then noisy student training? Many semi-supervised and active learning approaches could have been discussed, and some might potentially be more accurate than the proposed one. However, I do acknowledge one possible merit of the authors’ work, in generating a very lightweight baseline that requires no repeated training. However, this new method in a new problem setting needs clearer contextualization for sure.

---

> ### Author Response · Authors · 2022-08-02
> **Response for Reviewer oFCu [Cons 1-4]**
>
> **[Cons 1: use surrogate dataset]** Using a clean, publicly available surrogate dataset is a plausible option at the first glance, but many issues remain to be solved. 1) Most detection methods using clean datasets require at least one clean image (usually more than one) for each class [1,2,3,4,5,6]. This means the surrogate dataset should have very similar if not fully overlapped class categories. 2) Domain gap is inevitable, which may affect the effectiveness of detection methods.
> Even assuming the above roadblocks can be solved, such methods have limited effectiveness, as demonstrated by our following experiments. Specifically, we identify the DFG traffic sign dataset as a surrogate dataset to detect backdoor in GTSRB. Note that we process the DFG dataset using bounding box annotations to reduce the domain gap to GTSRB, and the results are provided in Table S2. The classes in the surrogate dataset have a large overlap with those of GTSRB and the target class is within the overlap classes. Even so, the existing detection methods NC [1] and STRIP [3] show unsatisfactory performance compared to ours. Hence the problem of detecting backdoor datasets cannot be solved by existing methods and our setting is not over-restricted.
>
> Table S2: Comparison of detection ability measured by AUROC between our proposed method and baseline methods NC and STRIP with the surrogate dataset.
> | Method | NC   | STRIP | Ours |
> |:--------:|:------:|:-------:|:-------:|
> | AUROC  | 0.46 | 0.55  | 0.88 |
>
>
> [1] Neural cleanse: Identifying and mitigating backdoor attacks in neural networks\
> [2] Tabor: A highly accurate approach to inspecting and restoring trojan backdoors in ai systems\
> [3] STRIP: A Defence Against Trojan Attacks on Deep Neural Networks\
> [4] DeepInspect: A Black-box Trojan Detection and Mitigation Framework for Deep Neural Networks.\
> [5] ABS: Scanning neural networks for back-doors by artificial brain stimulation\
> [6] Backdoor Scanning for Deep Neural Networks through K-Arm Optimization
>
> **[Cons 2: Performing self-supervised pre-training then noisy student training instead]** The proposed method assumes a trained model to be given and no re-training; it is hence orthogonal to the question “how to more robustly train a model over unreliable data '', which might indeed benefit from semi-supervised/active/noisy student learning. In fact, note that even SOTA pre-trained models can be vulnerable to backdoor attacks, and no evidence seems to support that transfer learning from a pre-trained model can defend against backdoor attacks better: see [7,8] et. al.
>
> [7] BadEncoder: Backdoor Attacks to Pre-trained Encoders in Self-Supervised Learning\
> [8] Backdoor Pre-trained Models Can Transfer to All
>
> **[Cons 3: hyperparameters -- threshold]** Our method shows good resilience to the hyperparameter within a range. Here we report the change in detection accuracy with respect to the threshold in Table S3. Actually, we use AUROC to measure the effectiveness of our method and the sensitivity of the threshold is implicitly considered in AUROC. Specifically, we change the threshold and depict the false positive rate and true positive rate, then generate the ROC curve and calculate AUROC, the area under the ROC curve.
>
> Table S3: Backdoor detection accuracy when changing threshold on various dataset-architecture combinations.
>
> | Threshold | 0 | 2 | 4 | 6 | 8 | 10 | 12 |
> | :-: | :-: | :-: | :-: | :-: | :-: | :-: | :-: |
> |BadNets, CIFAR-10, ResNet-20| 0.50 | 0.72 | 0.94 | 0.98 | 1.00 | 1.00 | 0.98 |
> |BadNets, CIFAR-10, AlexNet| 0.50 | 0.68 | 0.84 | 0.92 | 0.98 | 1.00 | 1.00 |
> |BadNets, GTSRB, ResNet-20| 0.50 | 0.56 | 0.80 | 0.84 | 0.88 | 0.72 | 0.68 |
> |CLA, CIFAR-10, ResNet-20| 0.50 | 0.76 | 0.94 | 0.96 | 0.94 | 0.90 | 0.86 |
> |WaNet, CIFAR-10, ResNet-20| 0.50 | 0.82 | 0.89 | 0.82 | 0.78 | 0.58 | 0.56 |
>
> **[Cons 4: hyperparameters -- layer choice]** We choose the last layers for detection because higher-layer features are well-known to be more class-discriminative.  And we also observe as layers go deep, feature separation in terms of sparsity characteristics between clean/backdoor becomes more obvious. Also, We conduct experiments to shuffle the first or random few layers (same total number of layers), and the results are provided in Table S4 which further endorses our analysis.
> Table S4: AUROC when using other layer choosing methods on various dataset-architecture combinations.
> | Attack  | Dataset   | Arch      | First | Random | Last (Ours) |
> | :-------: | :---------: | :-------: | :-----: | :------: | :-----: |
> | BadNets | CIFAR-10  | ResNet-20 | 0.53  | 0.64   | 1.000 |
> | BadNets | CIFAR-10  | AlexNet   | 0.46  | 0.86   | 0.997 |
> | BadNets | GTSRB     | ResNet-20 | 0.51  | 0.57   | 0.880 |
> | CLA     | CIFAR-10  | ResNet-20 | 0.61  | 0.5    | 0.997 |
> | WaNet   | CIFAR-10  | ResNet-18 | 0.53  | 0.48   | 0.954 |

---

> > ### Comment · Reviewer_oFCu · 2022-08-07
> > **Questions are solved. Raise the score to 7 from 6.**
> >
> > Many thanks for the detailed replies. The authors' explanation regarding the setting ([con1-2]) has well solved my main concern. New experimental results ([con3-4]) have eliminated my questions regarding the hyperparameters. I would like to raise the score to 7.

---

> > > ### Author Response · Authors · 2022-08-09
> > > **Thanks to reviewer oFCu**
> > >
> > > Dear Reviewer oFCu,
> > >
> > > Many thanks for the positive assessment of our work! We appreciate the helpful suggestions from reviewer oFCu.
> > >
> > > Best wishes,
> > >
> > > Authors

---

> ### Author Response · Authors · 2022-08-02
> **Response for Reviewer oFCu [Cons 5-7]**
>
> **[Cons 5: missing results in Table 3&4]** Thanks for pointing this out. Table 4 is the ASR (attack success rate) of recovered triggers based on models attacked by BadNets and Clean Label Attack (CLA) which includes input-agnostic trigger patterns causing all images misclassified. In contrast, the trigger of WaNet is input-dependent instead of fixed, which falls into a different category from BadNets and CLA.
>
> Meanwhile, the trigger recovery result for WaNet is in the Table A7 of the appendix, showing that we can still find an effective trigger pattern using the same recovery method to cause successful misclassification. We have followed the advice of reviewer oFCu and put TA & ASR results (table 3) for WaNet in Table A6 in the revised draft. We could put all WaNet results in the main text if you suggest.
>
> **[Cons 6: limited writing quality]** Thanks for pointing it out! The revisions are highlighted in blue color in the updated PDF file, and we would be happy to polish our writing further and make specific changes if you might suggest. We are also happy to see that other reviewers seem to appreciate our clarity, e.g. Reviewer AHdc comments as “The whole paper is easy to follow, and the main contributions are introduced clearly.”
>
> **[Cons 7: Inappropriate expression]**  Thanks for the advice! We will change the term to ‘feature differentiation’ in the revised version.

---

### Official Review · Reviewer_AHdC · 2022-07-12

**Rating:** 7
**Confidence:** 4
**Soundness:** 3 good
**Presentation:** 4 excellent
**Contribution:** 4 excellent

**Summary:**

 In this paper, the authors introduce and investigate a new backdoor detection setup without access to clean datasets. An effective detection algorithm is proposed by contrasting representations shifts before and after randomized channel shuffling of clean and trigger features. Moreover, by leveraging the proposed shuffly trained backdoor models, the performance of trigger recovery is substantially boosted. Extensive experiments validate the effectiveness of our proposal.

**Questions:**

Cons:

- Can we illustrate the trigger feature? It would be helpful to understand this key concept.

-  The trigger feature lacks a good mathematical definition, making further theoretical analysis hard.

- What is the false alarm rate of the proposed method? Namely the type I error in the field of hypothesis testing.

- Can the proposed method be used for adversarial data detection? Is that possible to distinguish between two attacks?

**Limitations:**

no potential negative societal impact of their work.

**Strengths And Weaknesses:**

Pros:

+ This paper is novel and introduces an interesting finding for backdoor attack detection. Specifically, it is the first time to try to detect such attacks without any clean data. Since we do not need clean data here, the proposed method can be used for many scenarios compared to previous methods.

+ The whole paper is easy to follow, and the main contributions are introduced clearly. The trigger feature hypothesis is very interesting and worthy to study more in the future.

+ The authors raise that the unreliability of training data source has become a serious security threat. They find that clean image features and trigger features are different in terms of their activation distribution in channels, as well as their sensitivity to channel re-ordering.

+ Utilizing trigger features’ unique characteristics, this paper proposes an effective, generalizable approach with minimal overhead based on channel shuffling to help data users examine datasets. The authors randomly shuffle weights of convolutional layers at the channel level, revealing the secretly hidden trigger features. This paper also uses the proposed shuffling method to retrain the models, helping generate more effective triggers.

+ Extensive experiments are used to verify the proposed methods.

Cons:

- Can we illustrate the trigger feature? It would be helpful to understand this key concept.

-  The trigger feature lacks a good mathematical definition, making further theoretical analysis hard.

- What is the false alarm rate of the proposed method? Namely the type I error in the field of hypothesis testing.

- Can the proposed method be used for adversarial data detection? Is that possible to distinguish between two attacks?

---

> ### Author Response · Authors · 2022-08-02
> **Response for Reviewer AHdC**
>
> **[Cons 1: Illustration of trigger feature]** Thanks for the advice! We generate $5\times 5$ gray-scale triggers [1] or RGB triggers [2] and place them at the lower right corner of 10\% of training samples to get backdoored datasets. The trigger patterns are provided in Appendix Figure A2.
>
> **[Cons 2: lack of definition of trigger feature]** We follow the trigger definition in [3] which is widely adopted. For completeness, we have  added the definition and discussion in the revised draft A1.3 as below:
>
> *“[3] defined a generic form of trigger injection. Let $\boldsymbol{x}_i $
> be an image sample and $\Delta$ denote a trigger pattern. The trigger is stamped on the image sample at a specific region characterized by a binary mask $\texttt{M}$. We say the resulting image is a backdoor image, denoted by $\boldsymbol{x}_i'$. Formally, the injection process is formulated as follows, where $\odot$ is the point-wise multiplication operator:
> $\boldsymbol{x}' = \boldsymbol{x} \odot(\mathbf{1}-\texttt{M})+\texttt{M} \odot \Delta$“*
>
> We agree that further theoretical study would be interesting, and some prior works have leveraged sparse feature models to formally study other relevant problems such as OoD detection [4]; that seems promising and we plan to explore this similar vein. However, a theoretical study falls out of the central scope of the current work, whose aim is to report an intriguing empirical finding and then utilize it for developing a new effective algorithm.
>
>
> **[Cons 3: False alarm rate of our method]** Thanks for pointing it out! In this work, the false alarm rate is the percentage of benign models that are wrongly identified as backdoor models. We provide full results in Table 1, and the false alarm rate can be calculated by (1- #benign/#total models), where the ‘benign’ column refers to the percentage of correctly classified models in all benign models. Our method has an average false alarm rate of 6.4% (1-117/125).
>
> **[Cons 4: Whether the method can be used for adversarial data detection]** Our method can potentially apply to adversarial detection due to intriguing similar properties between backdoor data and adversarial data. Specifically, the main foundation of our work is the hypothesis that trigger features are sparsely encoded in only a few channels. Such sparsity characteristic is also observed in adversarial data: for example, based on different sparse activation patterns of benign and adversarial samples, prior work [5] leverages the activation path feature to differentiate the two. However, their method requires access to clean data as references while ours does not. Given such an intriguing coincidence, we would love to explore, in our future work, whether the feature separation property can be a general observation of various attacks or distribution shifts.
>
> [1] BadNets: Identifying Vulnerabilities in the Machine Learning Model Supply Chain\
> [2] Hidden trigger backdoor attacks\
> [3] Neural cleanse: Identifying and mitigating backdoor attacks in neural networks\
> [4] DICE: Leveraging Sparsification for Out-of-Distribution Detection[5] Ptolemy: Architecture Support for Robust Deep Learning

---

> > ### Comment · Reviewer_AHdC · 2022-08-03
> > **Concerns solved, no further questions**
> >
> > I think the authors have solved my concerns in the letter.
> > con1-3 are solved by adding more experiments/explanations, which is good.
> > The authors provide a possible solution for con4.

---

> > > ### Author Response · Authors · 2022-08-06
> > > **Thanks to Reviewer AHdC**
> > >
> > > Dear Reviewer AHdC,
> > >
> > > Many thanks for all the helpful comments and very positive assessment. We really appreciate reviewer AHdC for increasing our score.
> > >
> > > Best wishes,
> > >
> > > Authors

---

### Official Review · Reviewer_raZy · 2022-07-12

**Rating:** 6
**Confidence:** 3
**Soundness:** 3 good
**Presentation:** 3 good
**Contribution:** 3 good

**Summary:**

This paper proposes the backdoor defense even without a clean data set. Specifically, after randomly shuffling the filters and comparing the changes in DNN output, the authors find there is an obvious difference between backdoored DNNs and benign DNNs. Futher They propose some metrics to detect whether a given model is backdoored or not. The experimental results demonstrate the effectiveness of the proposed method.

**Questions:**

For all-to-all backdoor attacks (samples from different classes are misclassified into different classes in the presence of the trigger pattern), does the proposed method still work?

**Limitations:**

Yes. This paper tries to provide a novel backdoor defense, which may mitigate the potential risk of backdoor attacks.

**Strengths And Weaknesses:**

**Strength**
1. When backdoor attacks bring practical threats to deployed DNNs, it becomes urgent and important to explore the backdoor defense. This paper has a good try.
2. To the best of my knowledge, the proposed method is novel and intriguing. I have never seen such a method, i.e., shuffling the filters, in backdoor defense.

**Weakness**
1. The method seems only to support the all-to-one attack, that is, all samples are misclassified into the same class in the presence of the trigger pattern.
2. This paper is a bit difficult to read and follow.

---

> ### Author Response · Authors · 2022-08-02
> **Response for Reviewer raZy [Cons 1 and 2]**
>
> **[Cons 1: Only support the all-to-one attack]**  We clarify that our approach does not only support the all-to-one attacks but also applies to cases when infected classes are minorities which is common for many backdoor detection methods [1,2,3,4,5,6,7]. Specifically, as is discussed in Sec 4.4 (line 363-369), our method generates decent results (AUROC = 0.902) when **4 out of 10** classes are infected. Note that our method also supports partial backdoor attacks (line 370-374). We will clarify this point in the final draft.
>
> [1] Universal litmus patterns: Revealing backdoor attacks in cnns\
> [2] STRIP: A Defence Against Trojan Attacks on Deep Neural Networks\
> [3] ABS: Scanning neural networks for backdoors by artificial brain stimulation\
> [4] Neural cleanse: Identifying and mitigating backdoor attacks in neural networks\
> [5] DeepInspect: A Black-box Trojan Detection and Mitigation Framework for Deep Neural Networks\
> [6] Tabor: A highly accurate approach to inspecting and restoring trojan backdoors in ai systems\
> [7] Practical detection of trojan neural networks: Data-limited and data-free cases
>
>
> **[Cons 2: Hard to read and follow]** Thanks for pointing it out! The revisions are highlighted in blue color in the updated PDF file, and we would be happy to polish our writing further and make specific changes if you might suggest. We are also happy to see that other reviewers seem to appreciate our clarity, e.g. Reviewer AHdc comments as “The whole paper is easy to follow, and the main contributions are introduced clearly.”

---

> > ### Comment · Reviewer_raZy · 2022-08-09
> > **Thank you for the reply**
> >
> > I appreciate the reply by the authors, which addresses my concerns. I also read the informative discussion between the authors and other reviewers. I have to admit that the authors indeed proposed a novel and interesting method, and the results are impressive. Thus, I raise my rating from 5 to 6.

---

> > > ### Author Response · Authors · 2022-08-09
> > > **Thanks to reviewer raZy**
> > >
> > > Dear Reviewer raZy,
> > >
> > > Thank you for the recognition for the merits of our work. We really appreciate your valuable suggestions.
> > >
> > > Best wishes,
> > >
> > > Authors

---

### Author Response · Authors · 2022-08-02
**Summary of Revisions**

We thank all reviewers for their careful reading and constructive feedback on the writing quality. We have made revisions to the paper per feedback, which are highlighted in blue in the updated PDF file. Major revisions include:

1. We have included some methods mentioned by reviewer PjHg in the introduction and pointed out the difference in setting with our method.
2. Use ‘feature differentiation’ instead of ‘feature disentanglement’.
3. We have included some recent attack methods in related work.
4. We have clarified and compared our method with more existing methods without the need for clean data (but in different settings).
5. Clarify the trigger type of clean label attack.
6. Illustrate the trigger pattern used in the attack in the appendix.
7. Provide a mathematical definition of triggers in the appendix.

We have also fixed all typos and grammatical errors.

---

### Meta-Review · Area_Chair_szt7 · 2022-08-28

**Recommendation:** Accept
**Confidence:** Certain

**Metareview:**

This work proposes a channel shuffling as a way to distinguish between backdoor and clean examples, based on the hypothesis that trigger features are sparsely encoded and activated in only a few channels. Reviewers all agreed that is a pretty intuitive, yet effective method and that it had solid evaluations after the rebuttal.

Reviewer oFCu had the concern that this paper is entirely empirical and has no supporting theory. I think this is ok given the precedence of papers in this field and also the framing of security and privacy is a more practically oriented one anyway.

The most critical reviewer (PjHg) pointed out that the benchmarks and related work contextualization were severely lacking. After the rebuttal period, these concerns were mostly alleviated.

Given the strength of the evaluations and the novelty of the idea, I believe this paper should be accepted.

That said, please address the following for the camera ready:
Please improve the writing as this was brought up by several reviewers. There are a lot of grammatical errors.
Please improve the discussion of the limitations section of this detection method as suggested by reviewer PjHg.

**Award:**

No

---

### Decision · Program_Chairs · 2022-09-14

Accept